# Iron Metabolism in Cancer

**DOI:** 10.3390/ijms20010095

**Published:** 2018-12-27

**Authors:** Yafang Wang, Lei Yu, Jian Ding, Yi Chen

**Affiliations:** 1Division of Anti-Tumor Pharmacology, State Key Laboratory of Drug Research, Shanghai Institute of Materia Medica, Chinese Academy of Sciences, Shanghai 201203, China; yafangwang123@163.com (Y.W.); lyu@jding.dhs.org (L.Y.); 2University of Chinese Academy of Sciences, Beijing 100049, China

**Keywords:** iron homeostasis, cancer, epigenetic regulation, tumor microenvironment, iron manipulating strategies

## Abstract

Demanded as an essential trace element that supports cell growth and basic functions, iron can be harmful and cancerogenic though. By exchanging between its different oxidized forms, iron overload induces free radical formation, lipid peroxidation, DNA, and protein damages, leading to carcinogenesis or ferroptosis. Iron also plays profound roles in modulating tumor microenvironment and metastasis, maintaining genomic stability and controlling epigenetics. in order to meet the high requirement of iron, neoplastic cells have remodeled iron metabolism pathways, including acquisition, storage, and efflux, which makes manipulating iron homeostasis a considerable approach for cancer therapy. Several iron chelators and iron oxide nanoparticles (IONPs) has recently been developed for cancer intervention and presented considerable effects. This review summarizes some latest findings about iron metabolism function and regulation mechanism in cancer and the application of iron chelators and IONPs in cancer diagnosis and therapy.

## 1. Introduction

The past decade has been described as the “golden age” of iron metabolism, due to the discovery of new iron-related proteins and regulatory mechanisms [1]. Efforts have been made to decipher physiological and molecular function of iron in cancer development. Multiple iron metabolism-associated proteins have been proved to participate in malignant tumor initiation, proliferation, and metastasis. Compared to normal cells, tumor cells differ in the expressions or activities of many iron-related proteins. These alterations generally contribute to a relatively high level of intracellular iron availability and facilitate the functions of iron-dependent proteins, which are involved in numerous physiological processes including DNA synthesis and repair, cell cycle regulation, angiogenesis, metastasis, tumor microenvironment, and epigenetic remodeling [2,3]. Consequently, iron homeostasis modulations including iron depletion and iron metabolism-targeted treatments have exhibit potent and broad anti-tumor effect, which makes it a potential and largely undeveloped therapeutic target for cancer pharmacological therapy. Some iron chelators and IONPs have already been put into clinical evaluation for curing hematological malignancies and other cancer types, and have lately shown enormous potential in combination with traditional chemotherapy and emerging immunotherapy [4,5]. Nevertheless, there exists more to be mined about iron homeostasis regulation and its role in cancer physiology, and ways to make it glow in cancer treatment. In this review, we integrate some latest expounded iron metabolism pathways and its major physiologies associated with cancer progression, tumor microenvironment, and epigenetic regulation. We then summarize some novel iron modulators in development and iron chelators in combined therapy, which could provide new therapeutic options for cancer intervention.

## 2. Regulation of Iron Homeostasis in Cancer

As a trace element, iron is necessary for cell basic function and especially highly required for malignant cancer cells, in which some pivotal changes about iron import and output have been identified. Generally, iron in the systemic iron pool is bound to transferrin (TF). Then, iron-loaded TF forms complex with transferrin receptor 1 (TfR-1) on the cell plasma membrane, which is internalized by endocytosis [6,7]. Whereas cancer cells have some alterant pathways in maintaining cellular iron balance. In non-small-cell lung carcinoma cells (NSCLC), epidermal growth factor receptor (EGFR) is demonstrated to affect iron metabolism by directly binding and re-distributing TfR-1. EGFR inactivation reduces TfR-1 level on the cellular surface, engendering iron import decrease and cell cycle arrest [8]. CD133 (cluster of differentiation 133), the pentaspan stem cell marker and a marker of tumor-initiating cells in a number of human cancers, can also inhibit iron intracellular uptake by interacting with TfR-1 and implicating in its endocytosis, thus participating in iron metabolism [9].

In the endosome, Fe^3+^ is reduced to Fe^2+^ by iron reductase, mainly by some members of the metalloreductases six-transmembrane epithelial antigen of prostate (STEAP1-4) family [10,11]. STEAP1 and STEAP2 are highly expressed in various human cancer types, such as colon, breast, cervix, prostate, pancreas, bladder, ovary, testis, and Ewing sarcoma [12,13,14]. STEAP3 is overexpressed in malignant gliomas, and STEAP3 knockdown suppresses glioma cell proliferation, clonality and metastasis in vitro and tumor growth in vivo. STEAP3 induces cancer epithelial–mesenchymal transition (EMT) by activating STAT3-FoxM1 axis, promoting TfR-1 expression and thus elevating cellular iron content [15]. STEAP4 is activated under hypoxia condition and leads to mitochondrial iron imbalance, enhances reactive oxygen species (ROS) production, and increases the incidence of colitis-associated colon cancer in mouse models [16]. Several promising STEAPs-targeting strategies in cancer therapy include monoclonal antibodies (mAbs), antibody-drug conjugates, DNA and small noncoding RNAs (ncRNAs) vaccines [17,18]. Once Fe^3+^ has been reduced to Fe^2+^ in the endosome, it is transported across the endosome into the cytosol via divalent metal-ion transporter 1 (DMT1), Zrt- and Irt-like protein 14 (ZIP14) or ZIP8 [19,20]. DMT1 functions as a main iron transporter and pharmacological inhibition of DMT1 suppresses colon tumor growth by suppressing JAK-STAT3 signaling [21,22]. The iron obtained through DMT1 constitutes the cytoplasmic labile iron pool (LIP) in which iron is metabolically active.

Most iron in the active form is finally utilized in various physiological processes such as DNA synthesis, mitochondrial oxidative metabolism and cytoplasmic ferritin for storage. Ferritin is an iron-containing protein with multiple functions in iron delivery, cell proliferation, angiogenesis, and immunosuppression. Under the case of cancer, ferritin is detected in high concentration in plasma in many patients, and its higher level correlates with higher clinical tumor stage and poorer patients’ outcome [23,24]. Iron–sulfur biogenesis is another common form of iron utilization. NEET proteins belong to a novel iron–sulfur (2Fe-2S) protein family that regulate iron and redox homeostasis and are involved in cancer progression. It has been revealed that NEET proteins NAF-1 and mitoNEET can promote cancer cell proliferation and metastasis by increasing mitochondrial iron accumulation. They represent a key regulatory link among the maintenance of high iron and ROS level in cancer cells [25,26].

Excess iron that is not utilized or stored can be exported across the membrane through ferroportin (FPN), an only-known iron efflux pump cooperated with ferroxidases named hephaestin (HEPH) or ceruloplasmin (CP) to maintain cellular iron homeostasis [27,28]. FPN is dramatically suppressed in many cancer types [29]. FPN overexpression induces autophagy and activates p53 and its downstream target p21, thus causing cell cycle arrest and stress-induced DNA-damage in prostate cancer [30]. Reduced FPN level in triple-negative breast cancer cells (TNBC) stimulates proliferation and epithelial-mesenchymal transition (EMT) as indicated by increased E-cadherin and decreased N-cadherin, Twist and Slug expression [31]. Some metal elements have been reported to modulate FPN’s transport activity, such as Ca^2+^ and Cadmium (Cd) [31,32]. These findings enrich our knowledge of FPN in mediating iron output and are conducive to the strategy development of manipulating FPN therapeutically in cancer.

Iron homeostasis has been demonstrated to be regulated under different levels. On the cellular level, iron metabolism is predominantly under post-transcriptional control by the iron responsive element-iron regulatory protein (IRE-IRP) system [33,34,35]. Under iron-low conditions, IRPs bind to IREs of mRNAs encoding ferritin subunits, FPN, DMT1 and TfR-1. Binding stabilizes TfR-1 and DMT1 mRNAs, whilst inhibits ferritin and FPN translation, which leads to an elevation in iron uptake and availability and a reduction in iron storage and efflux [36,37,38]. On the systemic level, iron homeostasis is mainly governed by hepcidin, a key iron sensing and regulatory hormone. Hepcidin facilitates FPN degradation and thus prevents iron export from gut enterocytes, reticuloendothelial cells (macrophages) and hepatocytes into circulation [39]. Hepcidin synthesized by tumors or liver contributes to cancer proliferation and progression. Significant genetic variants in the BMP/Smad4/Hamp hepcidin-regulating pathway could help predict the outcomes in NSCLC patients under definitive radiotherapy [40]. Studies have shown that regulating hepcidin level to reduce iron availability in the neoplastic cells may be a novel strategy in the anticancer treatment [41]. As acknowledged in tumor cells, proteins those raise intracellular iron contents (TfR-1, DMT1, hepcidin) are extensively upregulated, whereas those lower iron levels (FPN, hephaestin (HEPH)) are downregulated. Otherwise, they are identified as promising predictors for the clinical prognosis of patients with breast, liver, lung, colon, brain, prostate, ovarian, gastric and pancreatic cancer, leukemia, and glioma [19,42,43,44,45].

## 3. Iron and Epigenetic Regulation in Cancer

### 3.1. Iron Plays an Important Role in Cancer Epigenetics

When exist in redox-active form, iron can catalyze Fenton-type reactions and produce highly reactive free radicals, able to oxidize and attack most cellular components [46]. Fe-S cluster contributes the active center of various enzymes that are necessary for mitochondrial oxidative metabolism and epigenetic modulation. Its biogenesis disruption results in decreased overall histone acetylation by decreasing levels of histone acetyltransferase ELP3 (elongator complex protein 3) and succinate dehydrogenase, increased DNA methylation by reducing the activity of DNA demethyltransferase DME, and increased acetylation of a-tubulin in the cytosol by elevating the tubulin acetyltransferase MEC17 [47,48]. Moreover, an unbiased genetic screen shows that maintenance of iron homeostasis is sufficient to stabilize facultative heterochromatin assembly and appropriate genome-wide gene control. Iron overload may reshape the genome and epigenome by disrupting p53-mediated DNA repair pathway and enhancing DNA hypomethylation, forming a feasible environment for transposons or transposable elements [49,50]. These results uncover the mechanism of iron in globally remodeling the genome in a dynamic way.

Iron depletion has been shown to induce global histone and DNA methylation changes in cancer cells without affecting expression levels of histone and DNA methyltransferases or demethylases, implying the influence on their enzymatic activity [51,52]. Iron- and 2-oxoglutarate (2-OG)-dependent oxidative histone demethylation mediated by JmjC family are such enzymes, whose dynamic processes have been linked to the epigenetic regulation of tumor formation and development [53,54]. For instance, the JmjC-domain-containing protein JHDM2A/KDM3A, that specifically demethylates histone H3 at lysine 9 (H3K9), is iron-dependent and regulates androgen receptor transcription in prostate cancer cells [55]. JARID1B/KDM5B, which selectively demethylates H3K4me3, serves as a good prognosis predictor in NSCLC, breast and ovarian cancer [56,57,58]. The enzymatic activities of the JmjC family are significantly weakened in the absence of α-ketoglutarate, Fe (II) or ascorbic acid, or in the presence of iron chelators. Moreover, ten-eleven translocation (TET) proteins are also Fe(II)- and 2-OG-dependent dioxygenases that oxidize 5-methylcytosine (5mC) to 5-hydroxymethylcytosine (5hmC), 5-formylcytosine (5fC), and 5-carboxylcytosine (5caC), thereby generating active DNA demethylation, which is linked to aberrant silencing of tumor suppressor genes in cancer [59,60]. The deprivation of cellular LIP using specific chelator blocks the Tet-mediated 5mC oxidation [61]. Thus, iron is directly participated in the nucleus to promote oxidative demethylation of correlative DNA and histone residues throughout chromatin, which has been revealed to control cancer EMT status by regulating the expression of related proteins in a dynamic manner [62].

### 3.2. Epigenetic Regulation of Iron Metabolism-Related Proteins

From the above, iron makes a difference to epigenetics, whilst iron homeostasis is under epigenetic regulation which may take diverse forms, such as DNA methylation, histone modification, post-transcriptional control and some transcription factors working as cooperators (Figure 1). The HAMP gene, which encodes hepcidin, is transcriptionally repressed and closely associated with the hypermethylated signature on the gene promoter region in human hepatocellular carcinoma [63]. Histone deacetylase 1 (HDAC1) is defined as a novel hepcidin suppressor by binding to SMAD4, effect of which is independent of BMP/SMAD1/5/8 signaling and without deacetylation of SMAD4 or histone-H3 on the hepcidin promoter [64]. Histone deacetylase 3 (HDAC3) and its cofactor NCOR1 can also inhibit hepcidin expression, involving reversible loss of H3K9ac and H3K4me3 at the hepcidin gene promoter [65]. Hepcidin is also regulated by the E4BP4/G9a/SOSTDC1/hepcidin pathway which causes cellular iron dysfunction and is an essential link in thyroid cancer growth [66]. Apart from hepcidin, there are other iron-metabolism-related proteins under epigenetic control being identified. FTH is manipulated through histone acetylation by MBD5 (a methyl-CpG binding protein) forming complex with histone acetyltransferase KAT2A (lysine acetyltransferase 2A) [67]. Another MBD family member, methyl-CpG binding protein 2 (MECP2) deficiency is reported to mediate brain iron metabolism by inducing oxidative stress [68]. Our previous study shows that ferroxidase hephaestin (HEPH) is repressed by G9a, a H3K9 methyltransferase, which forms complex with transcription factor YY1 and HDAC1, leading to cellular LIP increase and promoting breast cancer proliferation [69]. The activity of IRP1 is modulated by NAD-dependent deacetylase sirtuin-3 (mitochondrial SIRT3), thus affects cellular iron metabolism. SIRT3 loss increases ROS production, bringing about elevated IRP1 binding to IREs and increased TfR-1 expression as a consequence [70]. Mitochondrial ferritin (FTMT) appears regulated by a complex mechanism involving epigenetic events, such as histone de-acetylases recruitment and GC islands accumulation at its promoter, and interplay between transcription factors, such as CREB, SP1, and YY1 [71]. Nuclear factor erythroid 2-like 2 (NRF2) and myeloid zinc finger-1 (MZF-1) could impact cancer cell growth by transcriptionally regulating FPN, FTH, and FTL expression in prostate and breast cancer [72]. FPN transcription is inhibited by transcription factor BACH1 (Btb and Cnc Homology 1) and activated by NRF2 [73]. Deacetylase SIRT2 can deacetylate and repress NRF2 nuclear localization, reducing FPN expression and iron export, thus maintaining cancer cellular iron levels [74]. While BACH1 is involved in iron and heme regulatory system since it is inactivated through directly binding to heme in erythroblast and K562 cells [75].

A set of miRNAs that functions in RNA silencing and post-transcriptional regulation of gene expression have been demonstrated to be involved in the regulation of iron metabolism. According to the recent studies, some miRNA levels correlates negatively with iron intake. In vitro experiments indicate that enhanced miR-320 level can repress TfR-1 expression and inhibit cell proliferation [76]. miR-7-5p and miR-141-3p target 3'-TfR-1 IREs and downregulate TfR-1 mRNA and protein level [77]. miR-210, miR-200a, miR-152 can also repress TfR-1 expression [78,79,80]. In erythroid cells, miR-Let-7d is involved in the refined regulation of iron uptake by targeting DMT1-IRE isoform. They are all inversely correlated during erythroid differentiation of CD34^+^ cells, K562 and HEL cells [81]. miRNAs in some way can also modulate iron utilization and storage. miR-638 overexpression reduces FTH protein level in prostate cancer [82]. miR-200b inhibits FTH expression and increases cancer cells sensitivity to chemotherapy [83]. The Fe-S cluster assembly protein (ISCU1/2) is suppressed by miR-210, leading to Fe-S cluster biogenesis disruption and iron metabolism alterations [84]. Furthermore, the liver specific miR-122 directly targets HFE and HJV and contributes to the regulation of systemic iron homeostasis by decreasing hepcidin mRNA expression [85]. The iron-binding glycoprotein lactoferrin has been identified as a direct target of miR-214 in MCF-7 cells. Nevertheless, its receptor is post-transcriptionally regulated by miR-584 in Caco-2 cells [86,87]. Except for regulating iron uptake, utilization and storage, miRNAs also take part in iron export regulation. miR-20a inhibits FPN by binding to highly conserved target sites in its 3′-UTR (untranslated region) [88]. Reinforced expression of miR-20a may diminish iron efflux, contributing to intracellular iron retention, which favors lung cancer cell growth and proliferation. miR-485-3p overexpression can also repress FPN expression and give rise to elevated intracellular ferritin levels and LIP content [89]. To be concluded, iron metabolism and epigenetic control are closely interplayed and mutual restricted in cancer.

## 4. Role of Iron in Tumor Cell Biology

### 4.1. Iron in Cancer Cell Cycle and DNA Metabolism

Due to higher rates of proliferation and DNA synthesis, neoplastic cells have higher requirements of iron than normal cells. Functioning in various electron transfer systems, iron is indispensable in maintaining the activity of iron- or heme-containing enzymes. They refer to amounts family of ferrous iron dioxygenases and each possesses active site dependent on relatively labile iron, including enzymes involved in DNA replication and repair, Jumonji (JmjC) domain-containing histone demethylases involved in epigenetic modification and mitochondrial oxidases involved in respiratory complexes [90,91,92]. To be more detailed, the proteins participated in DNA synthesis and repair procedures include DNA polymerases, DNA helicases, and the small catalytically active subunit of ribonucleotide reductase (RRM2) [92,93]. Moreover, mammalian cells contain a RRM2 subunit which is p53-inducible (p53R2) and can be activated in case of DNA damage. Interestingly, p53R2 is vulnerable to iron depletion, precipitating it into a potential target for iron chelation therapy in tumors with wild-type p53 [91]. Meanwhile, it is known that ROS generated in the Fenton reaction by excess iron can attack DNA, causing mutations and damages, inactivating tumor suppressor genes or activating oncogenes. Furthermore, iron plays a crucial role in regulating cell cycle by affecting both the formation and activity of the cyclin proteins (cyclin A, B, D, and E) and cyclin-dependent kinase (CDKs) complexes. Intracellular iron depletion by chelators results in hypo-phosphorylation of the retinoblastoma protein (pRb), decreased expressions of cyclins, p21 and c-myc, thus causes G1/S cell cycle arrest particularly [94,95]. Nevertheless, other researches also show that IRP2 depletion give rise to induction of p15, p21, and p27, leading to prostate cancer cells accumulation in G0/G1 [96]. Collectively, iron presents a close relationship with DNA metabolism and cell cycle process.

### 4.2. Iron in Cancer Cell Demise

To keep cellular redox homeostasis, iron and ROS levels are both strictly manipulated. When the balance is disrupted, the cell may go into canceration or demise. Since its first demonstration in 2012, ferroptosis has been well characterized as a type of programmed cell death caused by accumulation of iron-induced lipid peroxidation and metabolic constraints, which is genetically and biochemically distinct from apoptosis, autophagy, necroptosis, and necrosis [97,98]. Ferroptosis has been reported to be able to inhibit some cancer types, such as hepatocellular carcinoma (HCC), pancreatic carcinoma, breast cancer, and prostate cancer [99]. The glutathione (GSH) redox system is pivotal to restrain ferroptosis under the case of impaired lipid metabolism. Inhibition of GSH synthesis or glutathione peroxidase 4 (GPX4) is sufficient to trigger ferroptotic cell death [100]. The Fe-S cluster biosynthetic enzyme NFS1 is also critical to prevent ferroptosis by restricting iron influx from intracellular stores, cooperating with GSH synthesis [101]. Furthermore, wild-type p53 negatively regulates the expression of the cystine importer SLC7A11, which increases cell sensitivity to ferroptosis. p53 can also suppress ferroptosis through transcription-dependent and -independent mechanisms, implying the bidirectional and context-dependent control of ferroptosis and oxidative stress by p53 [102]. Recent studies reveal that the ferroptotic agent-induced endoplasmic reticulum (ER) stress contributes to the cross-talk among ferroptosis and other types of cell death, such as apoptosis [103]. Ferroptotic and apoptotic agents interact through the PERK-eIF2a-ATF4-CHOP-PUMA pathway caused by ER stress response and effectively enhance each tumoricidal efficacy, implicating a novel combined therapeutic strategy for cancer [103]. Furthermore, since cancer stem cells (CSCs) are iron-rich and iron-dependent, ferroptotic agent salinomycin and its derivative, ironomycin, exhibit selective and potent therapeutic effect against breast CSCs by accumulating iron-mediated lysosomal ROS via Fenton reaction and causing ferroptosis [104,105]. These unprecedented findings manifest the druggability of remodeling iron homeostasis in the context of CSCs.

Ferritinophagy refers to the autophagic degradation of ferritin protein that reserves iron and maintains balance when iron is depleted [106]. The process is mediated by an autophagy cargo receptor named nuclear receptor coactivator 4 (NCOA4), which binds to ferritin heavy chain (FTH) in the autophagosome and delivers it for degradation in the lysosome, thus releasing iron for physiological demands [107]. It has been revealed that ferritinophagy evokes an iron-driven intra-lysosomal oxidative reaction, resulting in LIP upregulation and lipid peroxidation, and finally cell demise [108].

Another iron-oxy-related cell death is ferrosenescence. It is defined as follows: excess iron directly degrades p53 and blocks the p53-mediated DNA repair, causing genomic disintegration and DNA damage in the cell. Aside from genomic alterations, ferrosenescence generates epigenomic changes by inducing global DNA hypomethylation through upregulating DNA methyltransferase 3A and 3B (DNMT-3A and DNMT-3B) and remobilizing transposable elements through miR-29/p53 pathway [50,109]. The ferrosenescence-associated genome disintegration eventually leads to cell demise by ferroptosis or apoptosis.

### 4.3. Iron in Tumor Metastasis and Angiogenesis

Iron plays an important role in matrix degradation and cancer metastasis by stimulating or stabilizing some metalloprotease activities. Iron overload increases metalloprotease-2 (MMP-2) and metalloprotease-9 (MMP-9) activity in a dose-dependent manner partly through activation of AP-1 via ERK/Akt pathway [110]. A recent study shows that FPN overexpression attenuates the LIP and ROS production and inhibits EMT, as reflected by significantly decreased representative EMT markers, such as SNAIL1, TWIST1, ZEB2, and vimentin [31]. FTH overexpression also leads to a suppression of EMT, which suggests that labile iron is beneficial for tumor migration [111]. However, some studies indicate that iron may inhibit vascular endothelial growth factor (VEGF)-induced endothelial cell proliferation, migration, tube formation, and sprouting [112]. Moreover, iron deficiency significantly promotes VEGF expression by stabilizing hypoxia-inducible factor-1α (HIF-1α) [113]. Therefore, iron has two sides on tumor metastasis under different circumstances which needs further research.

The iron-regulated metastasis suppressor N-myc down-stream-regulated gene 1 (NDRG1) is a well-known metastasis suppressor that decreases metastases and improves patient prognosis in breast, prostate, pancreas, and colon cancer [114,115]. Another NDRG family protein, the Myc-repressed gene NDRG2, though not as extensively studied as NDRG1, has demonstrated tumor-suppressive functions in malignant carcinomas [116,117]. Similar to NDRG1, NDRG2 is also upregulated under iron depletion. NDRG2 can reduce the level of receptor gp130 and inactivate its downstream targets STAT3 and ERK1/2, which leads to decreased EMT and tumor metastasis [118].

More iron-related proteins participating in tumor metastasis and angiogenesis include follows: collagen lysyl hydroxylases (LH1-3), a Fe^2+^- and 2-oxoglutarate (2-OG)-dependent oxygenases can maintain extracellular matrix homeostasis and cell migration potential [119]; β2-microglobulin (β2-M) interacts with its receptor, hemochromatosis (HFE) protein, stimulates iron responsive HIF-1α signaling pathway and promotes cancer bone and soft tissue migration [120]; inflammatory mediator Lcn2 increases tube formation, cell migration, and angiogenesis in rat brain endothelial cells via iron and ROS-dependent mechanisms [121]. All these suggest a significant role of iron in tumor progression.

### 4.4. Iron in the Tumor Microenvironment

The tumor microenvironment refers to the extracellular matrix (ECM), other non-malignant cells such as immune cells, surrounding blood vessels, also signaling molecules and cytokines around the tumor cells. Cancer initiation and progression largely depend on extrinsic signaling from their cell niche. In the past several years, a more detailed understanding of the interaction between iron metabolism and tumor microenvironment has been addressed (Figure 2). Inflammatory stressors in the tumor microenvironment play a critical role in controlling iron metabolism and homeostatic pathway [122]. Elevated iron in cancer cells and pericarcinomatous compartments protect cancer cells from natural killer cell cytolysis by upregulating ferritin expression and by antagonizing tumor necrosis factor (TNFα)- and NO-induced cytotoxicity [123]. Immunologic factors like interleukin-6(IL-6) cause significant upregulation of hepcidin through IL-6-STAT3 pathway, results in intestinal iron uptake suppression and serum ferritin levels elevation [124]. The relationships among cancer, inflammation, and iron-related proteins such as ferritin could be complex, with ferritin either indicating iron overload that causes cancer, or indicating inflammation that causes cancer [125]. It has been revealed that the dysregulation of iron-related proteins in cancer cells, macrophages and lymphocytes are correlated with clinicopathological markers of poor patients’ outcome, such as hormone receptor absence and tumor metastasis presence, extending the meaning of iron homeostasis in the tumor microenvironment [126].

The mononuclear phagocyte system (MPS, including macrophages, monocytes, and their precursor cells) dramatically participate in maintaining iron homeostasis by recycling iron from hemoglobin of damaged or senescent erythrocytes [127]. Cytokines like interferon-γ (IFN-γ) and tumor necrosis factor (TNF) secreted by many cell types, including Th1 cells, natural killer T (NKT) cells, monocytes, and macrophages, increase DMT1 expression whereas decrease FPN level, thus resulting in iron sequestration in the MPS. Macrophages, cancer cell and T-lymphocytes uptake and reserve non-transferrin bound iron (NTBI) through non-transferrin-bound iron transporters such as ZIP14 and DMT1, functioning as circulating iron isoforms to avoid different tissues from iron-induced cytotoxicity [128,129,130]. What we known is that macrophages play the leading role in taking up, metabolizing, storing, and releasing iron. Classically activated macrophages (M1 macrophages) sequester iron by absorbing iron-loaded TF via TfR-1 or taking up Fe^2+^ via zinc transporters ZIP8 and ZIP14 [131,132]. In alternatively activated macrophages (M2 macrophages), the major component of tumor-associated macrophages (TAMs), consume of hemopexin-heme via CD91 or haptoglobin-hemoglobin via CD163 into endosomes as well as phagocytosis of senescent erythrocytes into erythrophagosomes lead to heme accumulation in the cytosol, which is another important source of iron [133,134,135]. Intracellular Fe^2+^ is oxidized to Fe^3+^ by ferroxidase ceruloplasmin (CP) and exported from M2 macrophages through FPN, then binds to TF efficiently. Iron efflux via FPN is blocked by hepcidin through degradation, thus increasing iron storage in intracellular ferritin. An alternative channel of iron export is secretion of ferritin-bound iron which is abundant in TAMs. It has been revealed iron-enriched status increases M2 phenotype marker Arg1 and Ym1 expression and promotes M2 polarization, whereas it represses M1 proinflammatory response [136,137]. Though a recent study has pointed out that iron overload induces macrophage polarization to a pro-inflammatory phenotype through promoting ROS production, enhancing p300/CBP acetyltransferase activity and increasing p53 acetylation [138], iron indeed modulates the inflammatory response outcome. Iron-loaded TAMs (iTAMs) infiltration is reported to correlate with tumor regression in NSCLC patients. Iron delivery system targeted to TAMs has been proven to be an effective adjuvant therapeutic strategy to reinforce anti-tumor immune responses [139].

From the foregoing, ferritin is rich in TAMs, which have recently been demonstrated to have critical roles in tumor progression and drug resistance [140]. Inflammatory cytokines can regulate the expression of ferritin on two levels: transcriptional level (principally H-ferritin, FTH) and translational level (FTH and L-ferritin, FTL) [141]. FTH is highly expressed in the melanoma patients’ serum and is correlated with increased circulating CD4^+^CD25^+^ regulator T cells, contributing to their immune functions [142]. Further studies indicated that the proliferation of T cells such as CD8^+^ T cells requires intracellular iron stored in FTH [143]. FTL gathered in cancer lesions to promote proliferation is potentially taken up from plasma via specific receptors, such as scavenger receptor class A member-5 (SCARA5) from the release of TAMs, particularly in response to pro-inflammatory cytokines [140,144]. The increased iron uptake reinforces the IL-6 paracrine loop between TAMs and breast cancer cells, leading to intensive de novo acquired chemo-resistance [145]. Lactoferrin (Lf), an iron-binding glycoprotein, has multiple functions in innate immunomodulation by modulating cytokines production of granulocyte macrophage colony stimulating factor (GM-CSF), IL-1, TNF, and IL-6 by macrophages, regulating natural killer cell activity, and inhibiting antibody synthesis and T cells maturation [146,147].

The innate immune protein Lipocalin 2 (Lcn2) has emerged as a critical iron regulatory protein under tumorigenic and inflammatory conditions. TNF-α, IL-17, and IL-1β secreted from TAMs can induce Lcn2 expression by activating NF-κB pathway, suggesting that Lcn2 is a type I acute phase protein [148,149,150]. Lcn2 deficiency is shown to accentuate spontaneous colitis and promote colonic tumorigenesis in IL-10-deficient mice [151]. Lcn2 can also be released into the extracellular matrix and promote iron internalization and sequestration through known receptors such as megalin, contributing to cancer cell survival and metastasis. Mechanically, Lcn2 stabilizes and binds to MMP-9, resulting in matrix degradation and tumor EMT [152]. TAMs can secret Lcn2 and elevate intracellular iron concentration in tumor cells via Lcn2 as transporter [153]. In the TAMs, Lcn2 can colocalize with lactoferrin and release pro-inflammatory cytokines into the microenvironment [154]. It can also be stored as monomers or homodimers in the neutrophil-specific granules [155]. These concomitant events present the tumorigenic and immunological effects of Lcn2, thus facilitating tumor growth and metastasis. Neutrophil gelatinase-associated lipocalin (NGAL), another kind of lipocalin which is strongly expressed in thyroid carcinomas promotes leukocytes recruitment in tumor microenvironment through increased intracellular iron uptake and, consequentky, more chemokines production [156]. Furthermore, NGAL forms a complex with MMP-9 like Lcn2 and increases its stability which is crucial in cancer cell invasion as well as response to chemotherapy [157].

Apart from Lcn2, Hepcidin is another important factor in the intricate relationship between iron metabolism and tumor microenvironment. The core axis of hepcidin control is the BMP-HJV-SMAD signaling pathway and makes it the major target for pharmacologic intervene in cancer [158,159]. Activation of JAK/STAT3 signaling by inflammatory stimuli IL-6 can also enhance transcriptional activity of hepcidin gene (HAMP) [160]. Tumor-associated fibroblasts induce hepcidin expression via paracrine IL-6-BMP signaling, and this induction facilitates breast cancer cells growth [161]. Whereas the immunophilin FKBP12 represses hepcidin expression by binding the BMP type I receptor ALK2 and blocking BMP-SMAD pathway in hepatoma cells [162]. These findings may pave the way for using hepcidin targeting as a novel treatment for iron homeostasis in tumor tissue and the tumor microenvironment.

## 5. Iron Manipulating Strategies in Cancer

### 5.1. Iron is a Target for Oncotherapy

Both iron and ROS are cautiously managed to maintain balance or to stabilize their functions, thus a potential kind of cancer therapy is focused on disrupting the redox homeostasis by introducing or eliminating iron in the cell. Iron overload presented as ferric ammonium citrate (FAC) or iron complexes, remarkably inhibits cell survival in various cancer types [5,163,164]. Previous studies have shown that FAC induces cytoplasmic vacuolation formation in an ATG5/ATG7-dependent manner, with elevated LC3-II (autophagic marker) [159]. The synthetic Fe^2+^-polypyridyl complexes are able to inhibit glioblastoma tumor growth, and significantly induce TRAIL-mediated cell apoptosis by stimulating p38 and p53 and suppressing ERK pathway [165]. Furthermore, the iron complexes significantly inhibit tumor growth in vivo through enhanced cell apoptosis without evident systematic toxicities as confirmed by histological and pathological analysis [165,166].

On the other hand, iron chelators that deplete cellular iron level by binding iron with high affinity have been shown to suppress the proliferation of aggressive tumors like neuroblastoma and breast cancer, and lead to G1/S cell cycle arrest and apoptosis, suggesting that iron-deprivation may be an promising therapeutic strategy [167,168]. A number of studies have revealed that iron chelation can affect the AKT, ERK, p38, TGF-β, STAT3, JNK, Wnt signaling, and autophagic pathways to consequently suppress tumor growth and metastasis [137,169]. In clinical use, there are three drugs—deferasirox (DFX), deferiprone (DFP), and deferoxamine (DFO)—being evaluated for cancer treatment. DFX may selectively target the NF-κB ­pathway and induce highly specific apoptosis in myeloid leukemia, hepatoma and mantle cell lymphoma cells [170]. DFP has been reported to inhibit prostate cancer cell and TNBC cells proliferation and migration by decreasing oxygen consumption rate (OCR) and impairing mitochondrial function [171,172]. Megadose of DFO treatment disturbs intracellular iron homeostasis, induces apoptosis and represses growth in breast cancer cell lines [173]. Moreover, it has been suggested that iron chelation could impair not only tumor cells but also tumor microenvironment by affecting the polarized state of TAMs [174]. Taken together, high-dose iron chelators treatment instantaneously decreases LIP content to a level low enough to (1) cause remarkable disorder in cellular iron homeostasis; (2) induce DNA damage, cell cycle arrest and apoptosis; (3) modulate global histone methylation; (4) remodel cancer microenvironment; (5) inhibit cell growth and proliferation in both nonmetastatic and metastatic tumors.

Confronted with the efficacy of iron chelators as cancer therapy, a considerable body preclinical and clinical evidence points out that iron chelators also have nonnegligible toxic side-effects [175]. For example, iron depletion by DFO activates HIF-1α pathway and induces uPA and MMP-2 expression, which results in enhanced metastasis by degrading the extracellular matrix [176]. DFO treatment in the clinical trials has been reported to be able to increase the potent angiogenic factor VEGF level, leading to toxicities such as anemia and edema [177,178,179]. Therefore, novel types of iron chelators with fewer side-effects need to be discovered and more clinical evidences need to be confirmed and resolved. Further work is urgently required to find new iron chelators with maximized antitumor activity against a wide range of iron-overloaded cancer types, improve the oral activity, optimize the therapeutic schedule and clarify safety matters relating to iron deficiency anemia, panleukopenia, and edema. Therefore, the challenge is to design novel iron-scavenging agents those selectively kill cancer cells whilst leaving normal cells unaffected.

### 5.2. Combination Therapies and Novel Iron Modulators

Though the anti-tumor effect of iron chelators is very limited, iron deprivation in combination with chemotherapy has been certificated to improve each effectiveness without raising toxicity. DFX synergizes with standard chemotherapeutic agents such as doxorubicin, cisplatin, and carboplatin to suppress cell growth and cause apoptosis and autophagy in TNBC cells and other cancer types [180,181]. Moreover, in breast cancer patient-derived xenograft models, it has been reported that tumor recurrences are delayed by the combination without enhancing the side-effects of chemotherapies or impairing systematic iron homeostasis of the mice [182,183]. DFX can also synergistically repress pancreatic cancer cell proliferation with gemcitabine, a standard chemotherapy for pancreatic cancer, in vitro and in vivo [184].

Other than iron chelator combination strategy, iron and iron-based materials have been extensively studied for drug targeting and diagnostic applications [185,186]. Researchers have paid particular attention to iron oxide nanoparticles (IONP) because of their superparamagnetic properties which can be used for diagnosis and treatment. These nanoparticles engender cytotoxicity and genotoxicity in cancer cells via increasing ROS generation, oxidative stress, DNA damage, chromosomal condensation, and caspase-3 activity [187,188,189]. Recent results indicate that the photothermal effect of IONP can cause autophagy of cancerous cells in a laser dose-dependent manner. Co-treatment of IONP and autophagy inhibitor under laser exposure may suppress the xenograft tumor growth and facilitate LC3 production and TUNEL signaling, demonstrating a promising combination therapeutic method of INOP agents and autophagy modulators [190,191]. Moreover, IONP can function as optimal delivery intermediaries. Superparamagnetic iron oxide (SPIO) provide a potential siRNA delivery system, which accumulate easily in orthotopic tumor tissues, and avoid serum nuclease degradation [192]. Modified PEI-SPIO with the carrier Gal, which has a specific receptor on HCC cells, can target the c-Met siRNA specifically to tumor sites and effectively suppress tumor growth in a rat orthotopic model [192]. Taking this a step further, IONP is also applied to mediate the antibody-dependent cell-mediated cytotoxicity (ADCC). Multiple half chains of trastuzumab are conjugated onto magnetic iron oxide nanoparticles (MNP-HC) to form novel biological-active systems to improve target specificity and anti-HER2 therapeutic potential [193]. A novel combined use of IONP (superparamagnetic Fe_3_O_4_ nanoparticles) as a vaccine delivery platform and immune potentiator greatly promotes immune cells activation and cytokines production, stimulating potent cellular immune responses, and antigen specific CTL responses [188,194]. Macrophages and DC cells, functioning as the most professional antigen presenting cells, can be activated by Fe_3_O_4_-OVA nanoparticles and release diverse pro-inflammatory cytokines—including IL-6, interferon (IFN)-γ, and TNF-α in vitro—thus causing more potent immune responses [195,196]. Another evidence shows that IONP can strongly inhibit the biosynthesis metabolism of macrophages [197]. Furthermore, in recent years, the application of IONPs to visualize cell migration with magnetic resonance imaging (MRI) has been used clinically, showing its value in medical diagnosis [198,199]. All these attempts open a new therapeutic opportunity of iron modulators in different types of cancer and especially in drug-resistant tumors.

## 6. Conclusions

As far as we know, iron diversely functions in tumor initiation, progression, metastasis, and microenvironment. A large amount of iron in cancer cells is needed for the proliferation and progress. The expressions of numerous iron metabolism-related proteins are aberrantly regulated in malignant tumors, and a variety of signaling pathways and physiological processes are altered by iron in cancer, manifesting the essential roles of iron in cancer development. Overall, aberrant iron homeostasis is to some extent a hallmark of cancer. Thus, strategies based on remodeling iron homeostasis should offer promising choices for cancer therapy. Apart from diminishing intracellular iron content by iron chelators, targeting iron-associated proteins for drug delivery or impairing the redox status by elevating intracellular iron level have all been confirmed as feasible ways for cancer treatment. Despite amounts of iron-related signaling in cancer development have been illustrated in the existing studies, detailed understandings on the mechanisms of iron homeostasis maintenance, iron-associated proteins functions, and global effect on epigenetics and microenvironment involved of iron require further exploration. Last but not least, the efficiency and safety of the strategies based on iron metabolism regulation for cancer therapy still need more efforts to be improved.

## Figures and Tables

**Figure 1 ijms-20-00095-f001:**
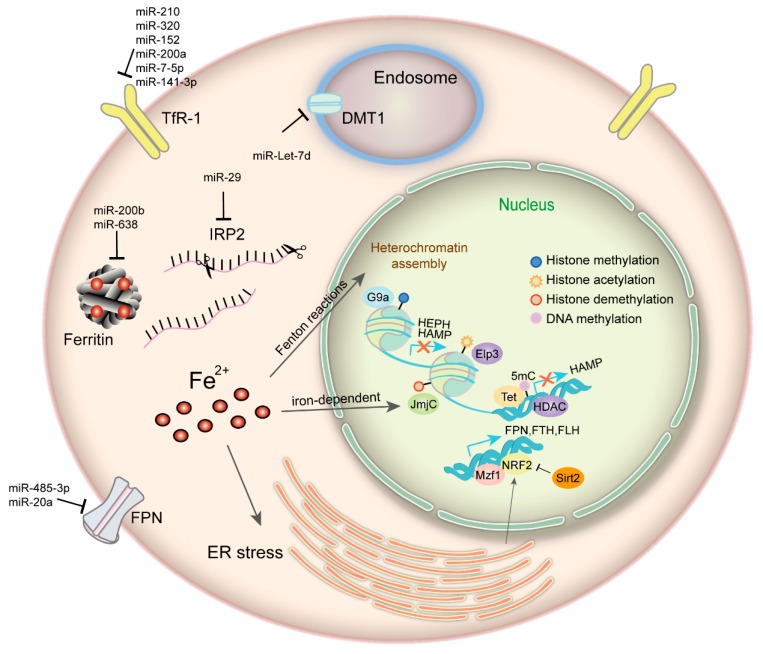
Iron and epigenetic regulation. Iron can modulate heterochromatin assembly mediated by Fenton reactions and induce global histone methylation changes through iron-dependent JmjC-domain-containing epigenetic modifying enzymes in cancer cells. Multiple miRNAs have been demonstrated to regulate iron metabolism-related proteins. DNA methylation, histone acetylation/methylation modification, and some transcription factors such as NRF2 and MZF-1 function corporately to maintain cellular iron metabolism in cancer. TfR, transferrin receptor 1; IRP2, iron regulatory protein; FPN, ferroportin; DMT1, divalentmetal transporter 1; ER, endoplasmic reticulum; HDAC, histone deacetylase; HEPH, hephaestin; Elp3, elongator complex protein 3; 5mC, 5-methylcytosine; TET, ten-eleven translocation protein.

**Figure 2 ijms-20-00095-f002:**
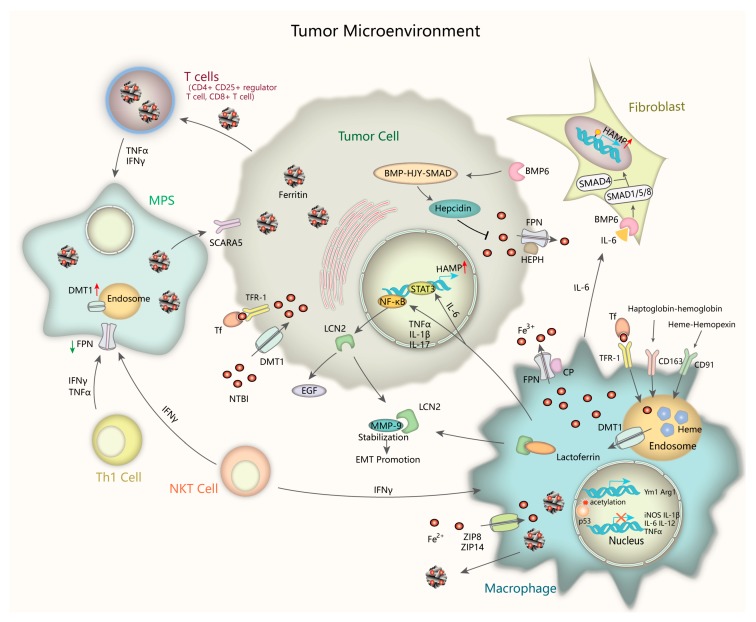
Iron handling in the tumor microenvironment. Tumor microenvironment compartments play a critical role in controlling iron metabolism. Inflammatory cytokines upregulate Lcn2 via NF-κB pathway. After releasing out of the cell, Lcn2 sequesters iron and stabilize MMP-9, promoting cell survival and matrix degradation leading to EMT. M2 macrophages are major sites of taking up, metabolizing, storing, and exporting iron. They supply iron to accelerate tumor growth by multiple transport pathways. Tumor-associated fibroblasts contribute to hepcidin induction via paracrine IL-6/BMP/SMAD signaling. Circulating T cells has accumulated H-ferritin to maintain proper immune functions. Th1 cells and NKT cells can secret cytokines like IFN-γ and TNF to the environment, which increase DMT1 whereas decrease FPN level, thus resulting in iron sequestration in the MPS. Tumor-associated fibroblasts induce hepcidin expression via paracrine IL-6-BMP-SMAD signaling. Lcn2, Lipocalin 2; NF-κB, nuclear factor kappa-light-chain-enhancer of activated B cells; MMP-9, matrix metalloproteinases-9; EMT, epithelial-mesenchymal transition; IL-6, interleukin-6; NKT, natural killer T cells; IFN—γ, interferon-γ; TNF, tumor necrosis factor; FPN, ferroportin; MPS, mononuclear phagocyte system.

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
