# Peer review of "Iron Metabolism in Cancer"

_ijms, 2018, doi:10.3390/ijms20010095_

Reviewer 1 Report

The manuscript by Wang et al. reviews the changes in iron regulation taking place in cancer cells, and updates data referring to the modulation of cancer cell growth and death through manipulation of iron levels. This is an exhaustive collection of references that should be useful to the researcher interested in iron cell biology and cancer.

However, in some parts of the text, the authors are too concise, on some occasions just mentioning the title of a paper, without further explanation. I would suggest that the authors revise the text and describe some sentences a little more, to make the manuscript more readable. Here follows a list of some unclear points.

1. Page 1, line 11, the authors state: By exchanging between oxidized and reduced forms, iron…

Since Fe3+ and Fe2+ are both oxidized forms of Fe, I would suggest changing the sentence to:

By exchanging between its different oxidized forms, iron…

2. Page 2, line 66, the authors state: After then, ferrous iron is transported across…

Since this sentence is a continuation of lane 55, maybe it is better to help the reader follow the reasoning by saying:

Once Fe3+ has been reduced to Fe2+ in the endosome it is transported across...

3. Page 2, line 72, the authors state: Most iron in the LIP is delivered to intracellular compartments such as the nucleus (for DNA…

To my knowledge, the cytosolic labile iron pool (LIP) is not transported to the nucleus. Please, correct this.

4. Page 2, line 75, the authors state: In the context of cancer, ferritin is detected in many patients and its higher level…

Low levels of serum ferritin can be detected in any healthy person. I understand that the authors wanted to say:

In the context of cancer, ferritin is detected in high concentration in plasma in many patients, and its higher level…

5. Page 2, line 80, the authors state: They represent a key regulatory link among the maintenance of high iron and ROS level in cancer cells, the transition of cell death and survival pathways [22, 23].

What does “the transition of cell death” mean?

6. Page 2, line 86, the authors state: Induction of FPN induces autophagy

What does “Induction of FPN” mean? Does it refer to the expression of the gene?

7. Page 3, line 102, the authors state: Nomograms incorporating significant SNPs in the BMP/Smad4/Hamp hepcidin-regulating…

This sentence is not clear, please explain.

8. Page 5, line 201, the authors state: Immunologic factors cause significant up-regulation of hepcidin through Interleukin-6…

If by immunologic factors they refer to LPS, it is better to state directly:

Lipopolysaccharide causes significant up-regulation of hepcidin through Interleukin-6…

9. Page 6, line 262, the authors state: Hepcidin is another important player in the daedal game…

What does daedal mean?

10. Page 7, line 285, the authors state: Fe-S cluster contributes the active center of various enzymes that compromise mitochondrial oxidative…

In homeostasis, these enzymes do not compromise mitochondrial oxidative metabolism. Probably the authors wanted to say:

Fe-S cluster contributes the active center of various enzymes that are necessary for mitochondrial oxidative…                            

11. Page 7, line 293, the authors state: Iron can destabilize the genome and epigenome by disabling p53-DNA repair pathway and enhancing hypomethylation, generating a favorable environment for transposons or transposable elements[114).

I might be wrong, but in reference 114 I was unable to find any mention to transposons. Please revise it.

Furthermore, English needs to be corrected by a native speaker.

Author Response

We appreciate the reviewers’ constructive suggestions to our manuscript, which greatly helped to enhance the quality of our manuscript. Our answers to the reviewer specific comments (repeated at the beginning of each paragraph and noted in bold) are:

1. Page 1, line 11, the authors state: By exchanging between oxidized and reduced forms, iron…

Since Fe3+ and Fe2+ are both oxidized forms of Fe, I would suggest changing the sentence to:

By exchanging between its different oxidized forms, iron…

 Answer: We are very grateful to your advice; we have made the amendment in the text.

2. Page 2, line 66, the authors state: After then, ferrous iron is transported across…

Since this sentence is a continuation of lane 55, maybe it is better to help the reader follow the reasoning by saying:

Once Fe3+ has been reduced to Fe2+ in the endosome it is transported across...

Answer: We have made the amendment in the text.

3. Page 2, line 72, the authors state: Most iron in the LIP is delivered to intracellular compartments such as the nucleus (for DNA…

To my knowledge, the cytosolic labile iron pool (LIP) is not transported to the nucleus. Please, correct this.

Answer: We are so appreciated to your suggestion, and corrected it in the text.

4. Page 2, line 75, the authors state: In the context of cancer, ferritin is detected in many patients and its higher level…

Low levels of serum ferritin can be detected in any healthy person. I understand that the authors wanted to say:

In the context of cancer, ferritin is detected in high concentration in plasma in many patients, and its higher level…

Answer: We have taken your advice in this version.

5. Page 2, line 80, the authors state: They represent a key regulatory link among the maintenance of high iron and ROS level in cancer cells, the transition of cell death and survival pathways [22, 23].

What does “the transition of cell death” mean?

Answer: The statement is indeed not clear enough, which has been deleted and stated as “They represent a key regulatory link among the maintenance of high iron and ROS level in cancer cells” in the text.

6. Page 2, line 86, the authors state: Induction of FPN induces autophagy

What does “Induction of FPN” mean? Does it refer to the expression of the gene?

Answer: Yes, we referred to the expression of the gene. To avoid any confusion, we have revised it in the text.

7. Page 3, line 102, the authors state: Nomograms incorporating significant SNPs in the BMP/Smad4/Hamp hepcidin-regulating…

This sentence is not clear, please explain.

Answer: We meant to state that significant SNPs in the BMP/Smad4/Hamp hepcidin-regulating pathway are found by nomograms incorporating method. Since it’s very confusing, we have made a relatively brief description in the text.

8. Page 5, line 201, the authors state: Immunologic factors cause significant up-regulation of hepcidin through Interleukin-6…

If by immunologic factors they refer to LPS, it is better to state directly:

Lipopolysaccharide causes significant up-regulation of hepcidin through Interleukin-6…

Answer: We are appreciated to your suggestion; one of the immunologic factors we refer to is Interleukin-6, which have been supplemented in the text.

9. Page 6, line 262, the authors state: Hepcidin is another important player in the daedal game…

What does daedal mean?

Answer: The word “daedal” means complex. To make it clear, we have revised in the text.

10. Page 7, line 285, the authors state: Fe-S cluster contributes the active center of various enzymes that compromise mitochondrial oxidative…

In homeostasis, these enzymes do not compromise mitochondrial oxidative metabolism. Probably the authors wanted to say:

Fe-S cluster contributes the active center of various enzymes that are necessary for mitochondrial oxidative…

Answer:  It has been in this version.                                  

11. Page 7, line 293, the authors state: Iron can destabilize the genome and epigenome by disabling p53-DNA repair pathway and enhancing hypomethylation, generating a favorable environment for transposons or transposable elements[114).

I might be wrong, but in reference 114 I was unable to find any mention to transposons. Please revise it.

Answer: We are sorry to make such a mistake. The correct references have been cited in the text.

Furthermore, English needs to be corrected by a native speaker.

Answer: Our manuscript has underwent extensive English editing, which made the expression more readable and clearer.

Reviewer 2 Report

The manuscript Explicate cancer in an iron wayby Yafang Wang et al., is an interesting and enough comprehensive review regarding the dysregulation of cellular iron metabolism in cancer. The authors report the findings of studies that concern the perturbations the iron metabolism in cancer cells.

In addition, the authors report therapeutic approaches for cancer based on altered iron metabolism in cancer cells.

The review is well written and is read with interest. The references are appropriate and up to date.

The suggestions to the authors are listed below:

- For an improved understanding it would be better to insert the paragraph 3. The role of iron in tumor cell biology before the paragraph 2. The regulation of iron homeostasis in cancer

- The authors should indicate in the text the Figures 1 and 2

- Pag. 3, line 107, it would be better indicate hephaestin (HEPH)

- In the References list, ref. 158 and ref. 160 are incomplete

- Pag 8, line 356, the ref. 146 does not seem to concern prostate cancer, please control it

- Pag 9, line 383, the ref. 154 does not seem to concern the iron overload on cell survival in cancer, please control it

- The authors should reference the following comprehensive review (Eid et al. 2017 Iron mediated toxicity and programmed cell death: A review and a re-examination of existing paradigms. Biochimica et Biophysica Acta (BBA) 1864:399-430).

Author Response

We appreciate the reviewers’ constructive suggestions to our manuscript, which greatly helped to enhance the quality of our manuscript. Our answers to the reviewer specific comments (repeated at the beginning of each paragraph and noted in bold) are:

- For an improved understanding it would be better to insert the paragraph 3. The role of iron in tumor cell biology before the paragraph 2. The regulation of iron homeostasis in cancer

Answer: We are so appreciated to your advice that we have changed the order of the paragraphs in the revised manuscript to make it more logical and comprehensible.

- The authors should indicate in the text the Figures 1 and 2

Answer: We have added the indication of Figure 1 and Figure 2 in the text.

- Pag. 3, line 107, it would be better indicate hephaestin (HEPH)

Answer: Thank you for your kindly suggestion. We have revised it in the text.

- In the References list, ref. 158 and ref. 160 are incomplete

Answer: We are terribly sorry to make such a mistake. The complete references have been cited in the text.

- Pag 8, line 356, the ref. 146 does not seem to concern prostate cancer, please control it

Answer: The correct reference has been cited in the text.

- Pag 9, line 383, the ref. 154 does not seem to concern the iron overload on cell survival in cancer, please control it

Answer: The correct reference has been cited in the text.

- The authors should reference the following comprehensive review (Eid et al. 2017 Iron mediated toxicity and programmed cell death: A review and a re-examination of existing paradigms. Biochimica et Biophysica Acta (BBA) 1864:399-430).

Answer: Thank you for your suggestion. The comprehensive review has been referenced in our revised manuscript.

Reviewer 3 Report

The Review describes the iron influence on cancer under different points of view. It is well written and offers a complete overview on the topic. The Authors should improve detailed description and bibliography on the role/signaling/biological functions elicited by iron toward tumor growth and progression. Moreover, the following references should be mentioned: Torti and Torti, Nat. Rev. Cancer 2013; Lui et al., Oncotarget 2015; Horniblow et al., Cancer Sci 2017; Chen et al., Oncogene 2015.

Author Response

We appreciate the reviewers’ constructive suggestions to our manuscript, which greatly helped to enhance the quality of our manuscript. Our answers to the reviewer specific comments are:

We are very grateful to your advice. Detailed description and bibliography on the role/signaling/biological functions elicited by iron toward tumor growth and progression have been improved in the revised version (such as Page 8, lines 322 to 326; Page 8, lines 355 to 358; all changes can be tracked). Moreover, the mentioned references have been referenced in the text.